# Targeted analysis of KRAS and CREBBP mutations uncovers a potential population-specific signature in thai patients with liver fluke-associated cholangiocarcinoma

Rehab Osman Taha[1], Wanna Chaijaroenkul[1,2], Papichaya Phompradit [ID][1], Kesara Na-Bangchang [ID][1,2]*

1 Graduate Program in Translational Bioclinical Sciences and Innovation, Chulabhorn International College of Medicine, Thammasat University, Pathum Thani, Thailand, 2 Center of Excellence in Pharmacology and Molecular Biology of Malaria and Cholangiocarcinoma, Chulabhorn International College of Medicine, Thammasat University, Pathumthani, Thailand

* kesaratmu@yahoo.com

## Abstract

Intrahepatic cholangiocarcinoma (iCCA) is an aggressive malignancy with limited therapeutic options and a poor prognosis. *Opisthorchis viverrini* (OV) infection is a major risk factor in endemic regions, particularly in Southeast Asia. However, the molecular mechanisms underlying iCCA development and progression remain incompletely understood. This study, as it is, is observational and demonstrates association rather than causation. This study aimed to characterize genetic alterations in key germline variants associated with cancer risk and prognosis, as well as components of the oxidative stress pathway, and to evaluate their associations with clinicopathological features in Thai iCCA patients. A cohort study was conducted involving 112 iCCA patients, 60 OV-infected individuals, and 156 healthy controls. Genetic alterations in *TP53*, *CREBBP*, *KRAS* (codons 12 and 13), *CDKN2A*, *IDH1*, and *GZMB* were analyzed by PCR and sequencing. Gene polymorphic co-occurrence and burden were assessed. Additionally, polymorphisms in the KEAP1–NFE2L2 oxidative stress pathway (*KEAP1 rs11085735*; *NFE2L2 rs6726395*, *rs6721961*, *rs4893819*) were analyzed in 50 iCCA patients. Associations with clinicopathological parameters, including metastatic status, tumor size, and tumor markers (CEA and CA 19−9), were evaluated using odds ratios (OR) and statistical analyses. *CREBBP* polymorphisms were significantly more frequent in iCCA patients (50.0%) than in OV-infected individuals (30.0%) and healthy controls (30.8%) (P = 0.003), with homozygous mutations conferring the highest cancer risk (OR = 6.43, 95% CI: 1.70–24.31). *KRAS* codon 13 polymorphisms were detected exclusively in iCCA patients (21.4%) and were absent in OV-infected individuals. In contrast, *TP53* polymorphisms were highly prevalent across all groups, with no significant differences, suggesting these variants may represent background genetic variation rather than tumor-specific drivers.

**Data availability statement:** All relevant data are within the manuscript and its Supporting Information files.

**Funding:** National Research Council of Thailand. The funders had no role in study design, data collection and analysis, decision to publish, or preparation of the manuscript.

**Competing interests:** The authors have declared that no competing interests exist.

Co-polymorphism analysis revealed that *TP53* and *CREBBP* alterations were the dominant genetic events, with most tumors harboring one or two mutations (mean gene polymorphism burden: 1.46±0.86). Further statistical modeling revealed significant clinicopathological associations. Binary logistic regression identified tumor size (P=0.029) and *TP53* mutation status (P=0.037) as significant predictors of metastasis. Notably, *TP53* wild-type status demonstrated a protective effect against metastasis (OR = 0.083, 95% CI: 0.007–0.950, P=0.045), and multivariable analysis confirmed *TP53* as an independent predictor of metastasis (P=0.037) after adjusting for sex, age, and sex-by-age interaction. Furthermore, ordinal regression identified metastasis as the primary predictor of advancing tumor stage (P<0.001), with tumor size showing a trending association (P=0.068). Evaluation of *CDKN2A* was limited by quasi-complete separation (adjusted OR = 0.000, P=1.000) due to sample size constraints. Analysis of the KEAP1–NFE2L2 pathway revealed limited genetic diversity in *KEAP1* but substantial polymorphic variation in *NFE2L2*. These polymorphisms showed minimal associations with clinicopathological features, suggesting a complex role of oxidative stress regulation in iCCA pathogenesis. The study identifies *CREBBP* and *KRAS* codon 13 polymorphisms as key genetic alterations enriched in iCCA, supporting their role as candidate germline variants and potential therapeutic targets. Polymorphism co-occurrence patterns indicate a relatively low mutational burden, with epigenetic dysregulation and oncogenic signaling representing central mechanisms in iCCA development. Further large-scale studies integrating tissue and circulating DNA analyses are warranted to validate these findings and identify clinically actionable biomarkers in iCCA.

## 1. Introduction

Cholangiocarcinoma (CCA) is a rare but highly aggressive malignancy arising from the epithelial cells of the bile ducts and accounts for approximately 20% of hepatobiliary cancers. The global incidence and mortality of CCA have increased over recent decades, with particularly high prevalence observed in Southeast Asia, including Thailand [1,2]. In northeastern Thailand, endemic infection with the liver fluke *Opisthorchis viverrini* (OV) represents a major public health concern. It is strongly associated with CCA development through chronic inflammation, oxidative stress, and DNA damage [3,4]. Early diagnosis of CCA remains challenging due to its asymptomatic nature in the early stages and the lack of reliable biomarkers, resulting in poor clinical outcomes. Although surgical resection remains the only potentially curative treatment, most patients present with advanced disease at diagnosis, limiting treatment options and resulting in high mortality rates [5,6].

The molecular pathogenesis of CCA is complex and involves genetic alterations, epigenetic dysregulation, metabolic reprogramming, and immune evasion mechanisms [5,7] dvances in genomic and transcriptomic profiling have identified key

oncogenic pathways and molecular subclasses of CCA, supporting the development of precision medicine approaches [8–10]. Among these genetic factors, inherited germline variants have garnered attention for their potential role in predisposing individuals to CCA. Unlike somatic mutations that arise during tumor progression, inherited variants may influence susceptibility to CCA from an early age, contributing to the disease's overall etiology. In particular, the dysregulation of glucose metabolism plays a central role in supporting tumor growth and survival. Glucose-6-phosphate dehydrogenase (G6PD), a key enzyme in the pentose phosphate pathway, provides NADPH, which is required for biosynthesis and maintenance of redox homeostasis in rapidly proliferating cancer cells [11,12]. Enhanced G6PD activity has been associated with tumor progression, chemoresistance, and poor prognosis in various cancers, highlighting its importance as a metabolic biomarker and therapeutic target [13]. However, G6PD was not detected in this study, which may limit its relevance in our analysis.

Inherited genetic variants in key oncogenes and tumor suppressor genes are critical in cholangiocarcinogenesis. Variants in the *TP53* gene, a central regulator of genomic stability and apoptosis, are among the most common alterations observed in CCA and are associated with tumor progression and poor clinical outcomes [14–16]. Furthermore, the *RAS* family of oncogenes (which includes *KRAS*, *HRAS*, and *NRAS*) plays a pivotal role in human oncogenesis. Functioning as binary molecular switches, RAS proteins cycle between an inactive GDP-bound state and an active GTP-bound state to properly relay extracellular signals from receptor tyrosine kinases to complex intracellular networks. Similarly, germline and somatic variants in *KRAS* -- the most frequently mutated RAS isoform in human cancers, particularly at hotspot codons 12 and 13, impair intrinsic GTPase activity and prevent the binding of GTPase-activating proteins (GAPs). This traps the KRAS protein in a constitutively active, GTP-bound conformation, leading to the continuous hyperactivation of downstream signaling pathways such as MAPK and PI3K. Consequently, these persistent signaling cascades promote uncontrolled cell proliferation, metabolic adaptation, and the evasion of apoptosis, directly contributing to tumor aggressiveness and poor prognosis [17–22]. Variants in *IDH1* represent a distinct molecular subtype of intrahepatic cholangiocarcinoma (iCCA), resulting in the production of the oncometabolite 2-hydroxyglutarate, which alters cellular metabolism and epigenetic regulation [23–25]. Additionally, inactivation of *CDKN2A* disrupts cell cycle control, facilitating uncontrolled proliferation [26], while germline mutations in *CREBBP* impair chromatin remodeling and transcriptional regulation, contributing to tumor development and progression [27,28]. In addition to these genetic factors, immune-related mechanisms also play important roles in CCA progression. Granzyme B (GZMB), a serine protease released by cytotoxic T lymphocytes and natural killer cells, is involved in tumor cell apoptosis and immune-mediated tumor surveillance [29–31]. While GZMB is one factor, it may not fully reflect changes in the tumor immune microenvironment, underscoring the need for further investigation of additional immune markers. Alterations in immune-related genes, such as *GZMB*, may reflect changes in the tumor immune microenvironment and serve as potential prognostic and therapeutic response biomarkers. Furthermore, oxidative stress response pathways involving nuclear factor erythroid 2-related factor 2 (NFE2L2/NRF2) and its regulatory protein Kelch-like ECH-associated protein 1 (KEAP1) play crucial roles in maintaining redox balance and promoting cancer cell survival by regulating antioxidant defense mechanisms and metabolic adaptation [32].

The present study was designed as a retrospective molecular analysis of archived biological specimens to investigate inherited genetic polymorphisms in intrahepatic cholangiocarcinoma (iCCA). Specifically, we evaluated germline variants in key cancer-related genes, including *TP53*, *KRAS* (codons 12 and 13), *IDH1*, *CDKN2A*, *CREBBP*, and *GZMB*, using PCR-RFLP and DNA sequencing. Additionally, the study investigated oxidative stress-related genes *KEAP1* and *NFE2L2* as potential markers of metabolic adaptation and tumor biology. These genetic profiles were compared among iCCA patients, individuals infected with OV, and healthy controls to assess their association with tumor development and progression. These findings may contribute to an improved understanding of the molecular mechanisms underlying cholangiocarcinoma and support the development of novel diagnostic and prognostic biomarkers for this aggressive malignancy.

## 2. Materials and methods

### 2.1. Study design and participants

This retrospective study used archived biological samples and clinical data (accessed date: 20 May 2024) from Thai patients originally collected at Sakonnakorn Hospital (Sakonnakorn Province) and Chonburi Hospital (Chonburi Province). The study protocols were approved by the Ethics Committee of Thammasat University and the Ministry of Public Health of Thailand (Approval No. MTU-EC-OO-4–150/63 and MTU-EC-OO-3–155/63). Based on a retrospective review of medical records, the subjects were classified into three distinct cohorts: Group 1 consisted of patients diagnosed with iCCA (n = 112), while Group 2 consisted of patients with OV infections who showed no signs of CCA (n = 60). Group 3 served as a control group, comprising healthy volunteers with no history of either CCA or OV infection (n = 156).

### 2.2. Chemicals and reagents

Genomic DNA was prepared using the Blood Genomic DNA Extraction mini kit (Favorgen, Pingtung, Taiwan), and the quality of extracted DNA and RNA was assessed using a NanoDrop spectrophotometer (ND-1000, Thermo Scientific, MA, USA). PCR reagents, including *Taq* DNA polymerase, Buffer, $MgCl_2$, and dNTPs, were obtained from Thermo Fisher Scientific Co. (MA, USA), while primer sequences for gene amplification were specially designed and synthesized. DNA amplification was carried out on a Biometra Thermal Cycler (Analytik Jena, Germany). For electrophoretic separation, Agarose powder was sourced from THEERA Co (Bangkok, Thailand), and NEOgreen DNA staining dye was acquired from Cell Gentek (Chungcheongbuk-do, Republic of Korea). Restriction enzymes (*BstUI*, *BstNI*, *HaeIII*, *Pvu1-HF*, *SacII*, *MnlI*, and *BseGI*) were purchased from New England Biolabs (MA, USA). Finally, the Gel/PCR Fragments Extraction Kit (Geneaid, Taipei, Taiwan) was used to purify genomic DNA.

### 2.3. DNA extraction and quality assessment

Genomic DNA was extracted from EDTA whole-blood samples collected from the three study groups using the Blood Genomic DNA Extraction mini kit, following the manufacturer's instructions. The quality and concentration of the extracted DNA were assessed using a NanoDrop spectrophotometer (ND-1000, Thermo Scientific, USA). The NanoDrop's micro-volume capability was particularly beneficial for these archival samples, given their low yields. Purity was determined by measuring the optical density (OD) ratio at 260 nm and 280 nm. Samples exhibiting an A260/A280 ratio between 1.8 and 2.0 were considered pure [33].

### 2.4. Genetic analysis

Genomic DNA from all subjects served as a template for the analysis of target genes using Polymerase Chain Reaction-restriction fragment length polymorphism (PCR-RFLP). Generally, PCR amplification was carried out in a 25 µL reaction volume containing 2 mM $MgCl_2$, 200 µM dNTPs, 1x PCR buffer, 0.25 µM of each primer, 2 µL of *Taq* DNA polymerase, and 3 µL of DNA template. The specific primer sequences, thermal cycling conditions, and annealing temperatures for *TP53*, *KRAS* (codons 12/13), *IDH1*, *CDKN2A*, *CREBBP*, *GZMB, KEAP1,* and *NFE2L2* are detailed in Table 1. Following amplification, PCR products were digested overnight at 37°C with the restriction enzymes listed in Table 2. Digested products were separated by electrophoresis on agarose gels (concentrations ranging from 2% to 3% depending on fragment size) stained with NeoGreen dye, and visualized under UV light. Genotyping was determined based on the banding patterns summarized below:

  *TP53:* Digestion with *BstUI* yielded bands of 416 bp (wild-type) or 161/263 bp (mutant).

  *KRAS:* Codon 12 digestion (*BstNI*) produced bands of 106/29 bp (wild-type) or 135 bp (mutant). Codon 13 digestion (*HaeIII*) produced bands of 85/48/26 bp (wild-type) or 85/74 bp (mutant).

  *IDH1: Pvu1-HF* digestion resulted in bands of 237/24 bp (wild-type) or 261 bp (mutant).

**Table 1. Primer sequences and protocol conditions for the PCR amplification of the investigated genes.**

| Target gene | Nucleotide Substitution | Amino Acid Substitution | Primer Sequence (5' –3') | PCR condition | Ref. |
|---|---|---|---|---|---|
| TP53 (rs1042522) | 215 C>G | 72 Arg>Pro | F: TGAGGACCTGGTCCTCTGACT<br>R: AAGAGGAATCCCAAAGTTCCA | 95°C for 5m, 35x (95°C for 30s, 58°C for 1m, 72°C for 30s), 72°C for 10 m | [34] |
| KRAS Codon12 (rs121913529) | 35 G>A | 12 Gly>Asp | F: CTGAATATAAACTTGTGGTAGTTGGACCT<br>R: TAATATGTCGACAAAACAAGATTTACCTC | 95°C for 5m, 45x (95°C for 60s, 55°C for 1m, 72°C for 60s), 72°C for 7 m | [35] |
| KRAS Codon 13 (rs121913527) | 38 G>A | 13 Gly>Asp | F: GTACTGGTGGAGTATTTGATGTGTATTAA<br>R: GTATCGTCAAGGCACTCTTGCCTAGG | 95°C for 5m, 40x (95°C for 60s, 50°C for 1m, 72°C for 30s), 72°C for 7m | [35] |
| IDH1 (rs121913499) | 394 C>T | 132 Arg >Cys | F: TGGGTAAAACCTATCATCATCGAT<br>R: TGTGTTGAGATGGACGCCTA | 95°C for 5m,38x (95°C for 30s, 50°C for 30s, 72°C for 30s), 72°C for 5 m | [36] |
| CDKN2A (rs3731249) | 442 T>C | 148 Ala>Thr | F: CTTCCTGGACACGCTGGT<br>R: AGTCTTCATTGCTCCGCAGT | 95°C for 5m,30x (95°C for 30s, 45°C for 30s, 72°C for 30s), 72°C for 7 m | [37] |
| CREBBP (rs3025684) | G>A | — | F: AGGGGAAACAACTCACCCTG<br>R: CTGGTCTTGTGGTTCCGTGT | 94°C for 5m,30x (94°C for 30s, 58°C for 30 s, 72°C for 30s), 72°C for 7 m | [38] |
| GZMB (rs8192917) | 143 C>T | 48 Gln>Arg | F: TCCCTAAGACAGGTATGCTC<br>R: GTGTTTCCAGGAGGGTGT | 94°C for 5m,36x (94°C for 30s, 60°C for 30 s, 72°C for 35s), 72°C for 10 m | [39] |
| GZMB (rs2236338) | 733 G>A | 245 Tyr>His | F: TCTCCCACATGTAGGCTGTG<br>R: GATGTGGTGCCTGAGAATGA | 94°C for 5m,30x (94°C for 30s, 58°C for 30 s, 72°C for 30s), 72°C for 7 m | [39] |
| KEAP1 (rs11085735) | 2882 C>A | 961 Asn>Ser | F:5´ CTC AGC CTC CCA AAG TCC CT<br>R: 5´ CTC CCA CGG CTG CAT CCA C 3´ | 95°C for 5m, 35x (95°C for 30s, 61°C for 30s, 72°C for 30s), 72°C for 5 m | [40] |
| NFE2L2 (rs6726395) | G>A | — | F:5'AAGGAGATCCCAGGATAAAAATC-3'<br>R:5'ACCAAGCAATGAAGCTGTCC-3' | 95°C for 5m, 45x (95°C for 60s, 55°C for 1m, 72°C for 60s), 72°C for 7 m | [41] |
| NFE2L2 (rs6721961) | G > T | — | FI:GGGCCCTGCCTAGGGGAGATGTGGACAACG<br>RI:TCAGGGTGACTGCGAACACGAGCTGCCAGA<br>FO:CACTTTACCGCCCGAGAATGGCGCCAGC<br>RO:CGTGGTGGCTGCGCTTTGGTGGGAAGAG | 95°C for 5m, 40x (95°C for 60s, 72.7°C for 30s, 72°C for 30s), 72°C for 7m | [42] |
| NFE2L2 (rs4893819) | T >C | | FI:TTAACAATTCAAGTTACTTATTAAAATGAC<br>RI:CTCATTGTCTACCTTCTCTGATGGCA<br>FO:AATTACTTGTAATTGAAGCAAGCTTCTT<br>RO:GAAAAGTGAAGGTTATTTCATTCAGTCT | 95°C for 5m,38x (95°C for 30s, 58°C for 30s, 72°C for 30s), 72°C for 5 m | [43] |

CDKN2A: *SacII* digestion yielded bands of 349/191 bp (wild-type) or 539 bp (mutant).

CREBBP: *MnlI* digestion produced bands of 101/44 bp (wild-type) or 145 bp (mutant).

GZMB: For *rs8192917*, *BsmAI* digestion showed bands of 328/92 bp (wild-type) or 420 bp (mutant). For *rs2236338*, *BsmFI* digestion showed bands of 272/128 bp (wild-type) or 400 bp (mutant).

KEAP1 *rs11085735*: *HinfI* digestion showed bands of 207/147 bp (wild-type) or 95 bp (mutant).

NFE2L2: For *rs6726395* *CviQI* digestion showed bands of 301/ 254 bp (wild-type) or 555 bp (mutant). For *rs4893819* (TETRA-ARMS PCR), results showed bands of 251 bp (wild-type) or 191 bp (mutant). For *rs6721961* (TETRA-ARMS PCR), results showed bands of 228 bp (wild-type) or 176 bp (mutant).

## 2.5. Agarose gel electrophoresis

PCR products and restriction fragments were separated on 1–3% agarose gels prepared in 0.5x TBE buffer and stained with Neogreen; the agarose percentage was adjusted according to the expected band size. For RFLP analysis, 10 µL of the PCR product was digested in a total reaction volume of 20 µL, containing 5 units of the specific restriction enzyme and the appropriate buffer. Following electrophoresis, fluorescent images of the gels were captured using a gel documentation system.

**Table 2. Restriction enzymes, conditions, and product size of the investigated genes.**

| SNPs | Restriction Enzyme | Incubation Condition | Fragment Size (bps) |
|---|---|---|---|
| TP53 (rs1042522) | BstUI | 37° C | CC: 416<br>GG: 161 + 263<br>CG: 416 + 263 + 161 |
| KRAS/Codon 12 (G12D) (rs121913529) | BstNI | 60° C | GG: 106 + 29<br>AA: 135<br>GA: 106 + 29 + 135 |
| KRAS/Codon 13 (G12D) (rs121913527) | HaeIII | 37° C | GG: 85 + 48 + 26<br>AA: 85 + 74<br>GA: 85 + 74 + 48 + 26 |
| IDH1 R132C (rs121913499) | Pvu1-HF | 37° C | CC: 237 + 24<br>TT: 261<br>CT: 237 + 24 + 261 |
| CDKN2A (rs3731249) | SacII | 37° C | TT: 349 + 191<br>CC: 539<br>CT: 539 + 349 + 191 |
| CREBBP (rs3025684) | MnlI | 37° C | GG: 101 + 44<br>AA: 145<br>GA: 145 + 101 + 44 |
| GZMB (rs8192917) | BsmA I | 37° C | CC: 328, 92<br>TT: 420<br>CT: 420, 328, 92 |
| GZMB (rs2236338) | BsmFI | 37° C | GG: 272 + 128<br>AA: 400<br>GA: 400 + 272 + 128 |
| KEAP1 (rs11085735) | HinfI | 37° C | AA: 207 + 147<br>AC: 207 + 147 + 112 + 95<br>CC: 147 + 112 + 95 |
| NFE2L2 (rs6726395) | CviQI | 60° C | GG: 301 + 254<br>AA: 555<br>GA: 555 + 301 + 254 |
| NFE2L2 (rs6721961) | HaeIII | 37° C | GG: 228<br>TT: 176<br>GT: 228 + 176 |
| NFE2L2 (rs4893819) | Pvu1-HF | 37° C | TT: 251<br>CC: 191<br>TC: 251 + 191 |

## 2.6. Statistical analysis

All statistical analyses were performed using SPSS Statistics version 26.0 (IBM Corp., Armonk, NY, USA). A two-sided $P$-value of < 0.05 was considered statistically significant.

Continuous variables, such as mutation burden, were summarized using the mean, standard deviation (SD), median, and range to characterize inter-patient variability. Categorical variables, including mutation status, genotype distributions, and co-occurrence patterns, were expressed as frequencies and percentages. Group comparisons for categorical variables, including differences in genotype and mutation frequencies among iCCA patients, OV-infected individuals, and healthy controls, as well as associations with clinicopathological parameters (e.g., metastatic status, tumor size, CEA, and CA 19−9 levels) were evaluated using Fisher's exact test. Pairwise comparisons between study groups (iCCA vs. healthy controls, iCCA vs. OV-infected, and OV-infected vs. healthy controls) were conducted to assess the disease risk associated with heterozygous and homozygous genotypes.

Inferential regression modeling was utilized to predict clinical outcomes based on demographic, clinical, and genetic factors. First, univariable binary logistic regression was performed to identify factors associated with metastasis. Predictor variables evaluated included sex, age, body mass index (BMI), tumor size, tumor stage, and specific genetic mutations (e.g., *TP53*, *KRAS13*, *CDKN2A*, and *CREBBP*). Subsequently, multivariable logistic regression models were constructed to determine the independent effects of these genetic markers on metastasis. Each genetic factor was evaluated while systematically adjusting for potential covariates, including patient sex, age, and the sex-by-age interaction term (Sex×Age). The magnitude of association for both univariable and multivariable models was expressed as crude and adjusted odds ratios (aORs) with their corresponding 95% confidence intervals (CIs). Finally, because tumor stage represents a naturally ordered progression, ordinal regression analysis (cumulative odds model) was performed. This analysis was utilized to evaluate the predictive value of demographic factors (sex, age), clinical characteristics (tumor size, metastasis status), and genetic mutations on advancing tumor stages.

## 3. Results

### 3.1. Demographic characteristics of study participants

Patients with iCCA were generally older, with ages ranging from 40 to 88 (mean±SD: 63.8±10.0) years. The OV group included individuals aged 32–70 (51.4±8.3) years, while the healthy control group included individuals aged 19–69 (38.6±12.2) years. Males (58.4%, n=66) were more common in the iCCA group than females (41.6%, n=47). The OV group showed a relatively balanced distribution between males (48.3%, n=29) and females (51.7%, n=31). In contrast, the healthy control group was predominantly female (82.5%, n=127), with substantially fewer males (17.5%, n=27).

### 3.2. Genetic polymorphisms frequency of candidate genes in iCCA patients

To characterize the molecular landscape of iCCA, targeted mutational analysis of key candidate genes (*TP53*, *KRAS* (codons 12/13), *IDH1*, *CDKN2A*, *CREBBP*, and *GZMB*) was performed on 112 iCCA samples. To comprehensively characterize the genetic landscape of the KEAP1–NRF2 pathway, a targeted analysis of four single-nucleotide polymorphisms (SNPs) was conducted in 50 patients with iCCA. These included one *KEAP1* variant (rs11085735) and three *NFE2L2* variants (*rs6726395*, *rs6721961*, and *rs4893819*). The prevalence and genotype distribution of all genes are summarized in Table 3.

   *TP53:* P53 emerged as the most frequently mutated gene in the iCCA cohort, with alterations detected in 73.2% (82/112) of analyzed samples. Stratification of these cases revealed that 58 (51.8% of the total cohort) harbored heterozygous mutations, while 24 (21.4%) exhibited homozygous mutations or loss of heterozygosity (LOH).

   *CREBBP:* CREBBP mutations represented the second most prevalent genetic alteration in the cohort, detected in 50.0% (56/112) of iCCA samples. The majority of these alterations occurred in the heterozygous state (46 cases, 41.1% of the total), while 10 cases (8.9%) demonstrated homozygous mutations.

   *KRAS:* KRAS mutations were identified exclusively at codon 13, occurring in 21.4% (24/112) of ICCA patients. Most mutations were heterozygous (22 cases, 19.6%), while only two cases (1.8%) exhibited homozygous alterations. Notably, no mutations were detected at the canonical codon 12, which represents the predominant hotspot in pancreatic and colorectal adenocarcinomas.

   *CDKN2A, IDH1, and GZMB:* In contrast to the frequent alterations observed in *TP53*, *CREBBP*, and *KRAS codon 13*, mutations in *CDKN2A* were exceedingly rare, detected in only one case (0.9%), representing a heterozygous alteration. No *IDH1* mutations were identified among the 112 analyzed samples (0.0%). Likewise, no *GZMB* variants were detected, including the two previously reported polymorphisms (*rs8192917* and *rs2236338*).

   *KEAP1 and NFE2L2:* The *KEAP1 rs11085735* variant exhibited very limited genetic diversity, with the wild-type CC genotype observed in 94.0% of patients. In contrast, the three *NFE2L2* polymorphisms (*rs6726395, rs6721961,* and

**Table 3. Prevalence and genotype distribution of the investigated genes in iCCA patients. Data are presented as numbers (n) and percentage (%) values.**

| Gene/SNP | Genotype | N | Percentage (%) |
|---|---|---|---|
| TP53 (rs1042522) | CC (Wild-type) | 30 | 26.8 |
| | GC (Heterozygous) | 58 | 51.8 |
| | GG (Homozygous variant) | 24 | 21.4 |
| CREBBP (rs3025684) | GG (Wild-type) | 56 | 50.0 |
| | GA (Heterozygous) | 46 | 41.1 |
| | AA (Homozygous variant) | 10 | 8.9 |
| KRAS 13 (rs121913527) | GG (Wild-type) | 88 | 78.6 |
| | GA(Heterozygous) | 22 | 19.6 |
| | AA (Homozygous variant) | 2 | 1.8 |
| KRAS 12 (rs121913529) | GG (Wild-type) | 112 | 100 |
| | GA (Heterozygous) | 0 | 0.0 |
| | AA (Homozygous variant) | 0 | 0.0 |
| CDKN2A (rs3731249) | TT (Wild-type) | 111 | 99.1 |
| | TC (Heterozygous) | 1 | 0.9 |
| | CC (Homozygous variant) | 0 | 0.0 |
| IDH1 (rs121913499) | CC (Wild-type) | 112 | 100 |
| | CT(Heterozygous) | 0 | 0.0 |
| | TT (Homozygous variant) | 0 | 0.0 |
| GZMB (rs8192917) | CC (Wild-type) | 112 | 100 |
| | CT (Heterozygous) | 0 | 0.0 |
| | TT (Homozygous variant) | 0 | 0.0 |
| GZMB (rs2236338) | GG (Wild-type) | 112 | 100 |
| | GA (Heterozygous) | 0 | 0.0 |
| | AA (Homozygous variant) | 0 | 0.0 |
| KEAP1 (rs11085735) | CC (Wild-type) | 47 | 94.0 |
| | AC (Heterozygous) | 3 | 6.0 |
| | AA (Homozygous) | 0 | 0.0 |
| NFE2L2 (rs6726395) | GG (Wild-type) | 8 | 16.0 |
| | AG (Heterozygous) | 26 | 52.0 |
| | AA (Homozygous variant) | 16 | 32.0 |
| NFE2L2 (rs6721961) | GG (Wild-type) | 1 | 2.0 |
| | GT (Heterozygous) | 29 | 58.0 |
| | TT (Homozygous variant) | 20 | 40.0 |
| NFE2L2 (rs4893819) | TT (Wild-type) | 2 | 4.0 |
| | TC (Heterozygous) | 39 | 78.0 |
| | CC (Homozygous variant) | 9 | 18.0 |

rs4893819) showed substantial allelic variation, with variant allele frequencies ranging from 48% to 69%, indicating greater genetic heterogeneity within the NRF2 pathway than within KEAP1 in this cohort.

### 3.3. Prevalence of genetic alterations among the three groups

Genetic alteration frequencies of the selected genes were compared across the study groups. *TP53* alterations were highly prevalent in all cohorts (73.2%, 78.3%, and 69.2% in iCCA, OV, and healthy individuals, respectively), with no

statistically significant difference among groups (P = 0.393). *KRAS* codon 13 polymorphisms were identified in 21.4% of iCCA patients, were absent in OV cases, and were present in 17.5% of healthy controls, demonstrating a highly significant difference (P < 0.001). In contrast, alterations in *KRAS* codon 12, *IDH1,* and *CDKN2A* were rare or undetectable across all cohorts and showed no significant differences. *CREBBP* polymorphisms were detected in 50.0% of iCCA patients, a frequency significantly higher than that observed in OV patients (30.0%) and healthy controls (30.8%) (P = 0.003). No *GZMB* variants (*rs8192917* or *rs2236338*) were identified in any group. Collectively, these findings indicate that *KRAS* codon 13 and *CREBBP* alterations are enriched in iCCA and may contribute to disease pathogenesis, whereas *TP53* alterations likely represent common background variants in this population, as shown in Table 4.

Further analysis demonstrated a strong and specific association between *CREBBP* polymorphisms and iCCA (Table 5). Both heterozygous and homozygous polymorphisms were significantly more frequent in iCCA patients than in healthy controls, with evidence of a gene-dosage effect indicating increased cancer risk with greater polymorphic burden. Heterozygous mutations were associated with a nearly two-fold increased risk of iCCA (OR = 1.97, 95% CI: 1.17–3.32, P = 0.015), while homozygous mutations conferred a substantially higher risk (OR = 6.43, 95% CI: 1.70–24.31, P = 0.006). Correspondingly, the wild-type genotype was less common in iCCA patients (50.0%) than in healthy controls (69.2%). Comparisons between iCCA and opisthorchiasis (OV) groups showed a similar pattern, with higher frequencies of both heterozygous (OR = 2.03, 95% CI: 1.02–4.03, P = 0.061) and homozygous mutations (OR = 7.50, 95% CI: 0.92–60.89, P = 0.065) in iCCA, although statistical significance was not reached. In contrast, *CREBBP* genotype distributions in OV-infected individuals were comparable to those in healthy controls, with no significant differences observed. Overall, these findings indicate that *CREBBP* polymorphisms are specifically associated with malignant transformation in iCCA rather than with OV infection alone, and that increasing mutational burden may contribute to greater cancer susceptibility.

**Table 4. Comparison of genotype distribution of *TP53, CREBBP, KRAS, CDKN2A, IDH1, GZMB, KEAP1,* and *NFE2L2* mutations in iCCA, OV, and healthy groups. Data are presented as numbers (n), total number of samples analyzed (N), and frequency (%).**

| Gene | CCA | | OV | | Healthy | | P-value |
|---|---|---|---|---|---|---|---|
| | (n/N) | Frequency (%) | (n/N) | Frequency (%) | (n/N) | Frequency (%) | |
| *TP53* | 82/112 | 73.2% | 47/60 | 78.3% | 108/156 | 69.2% | 0.393 |
| *KRAS 12* | 0/112 | 0.0% | 0/60 | 0.0% | 0/156 | 0.0% | 1.000 |
| *KRAS 13* | 24/112 | 21.4% | 0/60 | 0.0% | 27/154 | 17.5% | <0.001* |
| *CDKN2A* | 1/112 | 0.9% | 0/60 | 0.0% | 0/156 | 0.0% | 0.380 |
| *IDH1* | 0/112 | 0.0% | 0/60 | 0.0% | 0/156 | 0.0% | 1.000 |
| *CREBBP* | 56/112 | 50.0% | 18/60 | 30.0% | 48/156 | 30.8% | 0.003* |
| *GZMB (rs8192917)* | 0/112 | 0.0% | 0/60 | 0.0% | 0/156 | 0.0% | 1.000 |
| *GZMB (rs2236338)* | 0/112 | 0.0% | 0/60 | 0.0% | 0/156 | 0.0% | 1.000 |

\* Statistically significant difference among the three groups (Fischer's exact test).

**Table 5. Comparison of the distribution of *CREBBP* genotypes among the iCCA patients, OV patients, and healthy volunteers.**

| Comparison | Case Group | Control Group | Key Findings |
|---|---|---|---|
| iCCA vs. Healthy | iCCA (n=112) | Healthy (n = 156) | iCCA: significantly increased risk (Het: OR=1.97, P = 0.015; Hom: OR=6.43, P = 0.006) |
| OV vs. Healthy | OV (n = 60) | Healthy (n = 156) | OV: NO significant difference from Healthy (Het: OR=0.97, P = 1.000; Hom: OR=0.86, P = 1.000) |
| iCCA vs. OV | iCCA (n=112) | OV (n = 60) | iCCA: marginally increased risk vs OV (Het: OR=2.03, P = 0.061; Hom: OR=7.50, P = 0.065) |

### 3.4. Association between genetic alterations and clinicopathological characteristics

Overall, the clinicopathological correlation analysis revealed no definitive associations between the examined genetic alterations and metastatic status, tumor size, and tumor markers (CEA and CA 19−9). For tumor metastasis and tumor size, respectively, the analysis was based on 35 and 55 iCCA patients with documented metastatic status (25 with metastasis and 10 without). *TP53* alterations demonstrated a trend toward increased metastatic risk (OR = 3.86, P=0.123). An inverse relationship was observed between *KRAS* codon 13 polymorphisms and tumor size (P=0.064). For *KEAP1 rs11085735* and the *NFE2L2* variants *rs6726395* and *rs6721961*, none demonstrated significant associations with metastatic status or tumor size (P>0.05). For tumor markers, *NFE2L2 rs6721961* (GT+TT) and *NFE2L2 rs4893819* (TC+CC) polymorphisms showed significant association with increased levels of CA 19−9 [median: 50.5 vs. 13701.5 IU and 50.5 vs. 844 IU, for wild-type and polymorphic genes, respectively].

To further evaluate these relationships, regression analyses were performed. In the unadjusted binary logistic regression model, tumor size (P=0.029) and *TP53* mutation status (P=0.037) emerged as statistically significant predictors of metastasis. Specifically, *TP53* wild-type status was associated with a lower risk of metastasis compared with the mutated reference group (P=0.045, 95% CI: 0.007–0.950). In the multivariable model, *TP53* remained an independent predictor of metastasis (P=0.037) even after adjusting for patient age, sex, and the sex-by-age interaction. Furthermore, ordinal regression analysis used to predict tumor stage indicated that metastasis was the only highly significant factor (P<0.001), while tumor size showed a trend toward significance (P=0.068).

### 3.5. The patterns of co-mutations among key genetic alterations in iCCA

Patients were stratified according to distinct mutation patterns, and the number of altered genes *per* patient was determined to assess mutation burden and co-alteration tendencies (Table 6). *TP53* and *CREBBP* polymorphisms, either alone or in combination, represent the predominant genetic events in iCCA, affecting the majority of cases. *KRAS* codon 13 polymorphisms were rarely observed as isolated events and more commonly occurred alongside alterations in *TP53* and/or *CREBBP*.

### 3.6. Distribution of mutation burden among iCCA patients

To evaluate the overall mutational burden and inter-patient variability in genetic complexity, the number of mutations per patient was analyzed (Table 7). A significant proportion of iCCA patients exhibited at least one detectable polymorphism: 87.5% had at least one, while 12.5% had none. The largest group of patients (41.1%) had a single polymorphism, followed by 34.8% with two polymorphisms. A smaller subset of patients (11.6%) presented with three polymorphisms. The mean number of mutations per patient was 1.46 ± 0.86, with a median of 1.0 and a range of 0–3 mutations. This suggests moderate inter-patient variability and a distribution that skews toward fewer polymorphisms. Overall, more than half of the patients (53.6%) had one or fewer mutations, and nearly 90% (88.4%) had two or fewer polymorphisms.

**Table 6. Patterns of co-mutations of genetic alterations in iCCA patients. Data are presented as numbers and percentage (%) values.**

| Rank | Mutation Pattern | Number of Genes | Number of Patients | Percentage (%) |
|------|------------------|-----------------|--------------------|----------------|
| 1 | *TP53* | 1 | 33 | 29.5% |
| 2 | *TP53+CREBBP* | 2 | 29 | 25.9% |
| 3 | *CREBBP* | 1 | 12 | 10.7% |
| 4 | *TP53+KRAS 13+CREBBP* | 3 | 12 | 10.7% |
| 5 | *TP53+KRAS 13* | 2 | 7 | 6.2% |
| 6 | *KRAS 13+CREBBP* | 2 | 3 | 2.7% |
| 7 | *KRAS 13* | 1 | 1 | 0.9% |
| 8 | *TP53+KRAS 13+CDKN2A* | 3 | 1 | 0.9% |

**Table 7. Distribution of mutation burden among iCCA patients. Data are presented as numbers and percentage (%) values.**

| Number of Mutations | Number of Patients | Percentage (%) | Cumulative Percentage (%) |
|---|---|---|---|
| 0 | 14 | 12.5% | 12.5% |
| 1 | 46 | 41.1% | 53.6% |
| 2 | 39 | 34.8% | 88.4% |
| 3 | 13 | 11.6% | 100.0% |
| Mean±SD | 1.46±0.86 | – | – |
| Median (Range) | 1.0 (0-3) | – | – |

## 4. Discussion

This study provides a comprehensive molecular characterization of iCCA in a Thai cohort, with comparative analysis against OV-infected individuals and healthy controls. Our findings reveal distinct genetic alteration patterns that enhance understanding of the molecular pathogenesis of iCCA and distinguish tumor-associated germline variants from background genetic variation. The identification of *CREBBP* and *KRAS* codon 13 polymorphisms as significantly enriched in iCCA, together with characterization of mutation burden and co-occurrence patterns, provides important insights into the genetic architecture of this malignancy and has potential implications for prognostic stratification and targeted therapy [10,44,45].

### 4.1. Germline variants and molecular pathogenesis of iCCA

Our findings suggest that *CREBBP* and *KRAS* polymorphisms may serve as valuable biomarkers for identifying individuals at higher risk of developing iCCA. This could lead to earlier screening and intervention, particularly in populations with high CCA prevalence, such as those with chronic OV infections. The comparative analysis across iCCA, OV-infected, and healthy control groups was critical for distinguishing true oncogenic germline variants from germline polymorphisms. The increased frequency of *CREBBP* polymorphisms in iCCA patients supports its potential role as a candidate tumor suppressor gene, which could be leveraged to develop targeted therapies or preventive measures [46].

A critical challenge in the clinical management of endemic iCCA is predicting the transition from chronic OV infection to overt malignancy. In our cohort, comparing the genetic profiles of OV-infected patients and iCCA patients revealed distinct mutational timelines. Notably, *KRAS* mutations were conspicuously absent in the OV group but present in the iCCA group. Chronic OV infection is known to initiate carcinogenesis primarily through mechanical damage, chronic inflammation, and the generation of reactive oxygen species, which drive early genomic instability. The absence of *KRAS* alterations in the pre-malignant OV phase suggests that *KRAS* is not an initiating event, but rather a late-stage driver mutation required for full malignant transformation [19]. Consequently, the sudden emergence of a *KRAS* mutation in the circulating cfDNA of an OV-infected patient could serve as a high-confidence, predictive biomarker indicating active transition to iCCA. This highlights the potential of longitudinal ctDNA monitoring in high-risk, OV-endemic populations to catch iCCA at its earliest, most intervention-amenable stage. Interestingly, our *in silico* analysis of the TCGA-CHOL cohort revealed that the *CREBBP* and *KRAS* co-mutation is relatively rare in that dataset, thereby preventing robust survival analysis. However, this is a highly anticipated and biologically significant finding: the TCGA cohort is predominantly composed of Western, non-fluke-associated iCCA patients, whereas our cohort consists of Thai patients with iCCA in an OV-endemic region. The unique enrichment of this *CREBBP*/*KRAS* signature in our dataset strongly suggests it may be a distinct molecular hallmark of fluke-associated cholangiocarcinogenesis [47].

In contrast to our findings in the Thai population, previous comprehensive genomic profiling in Chinese cohorts has identified different dominant driver pathways, though *KRAS* remains a significant driver [48]. This divergence likely stems from different etiological backgrounds, as Chinese and Japanese iCCA cases are predominantly associated with viral

hepatitis or Clonorchis sinensis. In contrast, our cohort is strictly exposed to endemic OV. Consequently, the *CREBBP/KRAS* mutational profile observed here may represent a unique, population-specific signature for OV-endemic iCCA.

In particular, the dysregulation of glucose metabolism plays a central role in supporting tumor growth and survival. Glucose-6-phosphate dehydrogenase (G6PD), previously identified as a key metabolic enzyme, was not detected in our study, suggesting its role may be less significant in this cohort. Future studies should explore alternative metabolic pathways that are active in iCCA, potentially identifying new therapeutic targets [49].

Inherited genetic variants in key oncogenes and tumor suppressor genes are critical in cholangiocarcinogenesis. Variants in the *TP53* gene, a central regulator of genomic stability and apoptosis, are among the most common alterations observed in CCA. Our study emphasizes the need for continued monitoring of *TP53* variants in clinical settings, as they may indicate tumor progression and poor clinical outcomes. Similarly, germline variants in *KRAS*, particularly at codons 12 and 13, can promote uncontrolled cell proliferation by constitutively activating downstream signaling pathways such as MAPK and PI3K. These findings highlight the potential of KRAS-targeted therapies to improve prognosis and treatment outcomes for patients with iCCA [50,51].

## 4.2. Importance of immune-related mechanisms

Immune-related mechanisms also play important roles in CCA progression. Granzyme B (GZMB) is involved in tumor cell apoptosis and immune-mediated tumor surveillance. While GZMB alone may not fully represent the changes in the tumor immune microenvironment, its presence could indicate a more active immune response in patients, which may correlate with a better prognosis [29,30,39]. Thus, understanding the immune landscape in iCCA could inform future immunotherapeutic strategies and enhance patient management. Furthermore, oxidative stress response pathways involving NFE2L2/NRF2 and KEAP1 play crucial roles in maintaining redox balance and promoting cancer cell survival. Targeting these pathways may offer new avenues for treatment, especially when conventional therapies fail.

## 4.3. Clinical implications and future directions

The identification of *CREBBP* and *KRAS* polymorphisms as significantly enriched in iCCA has important implications for precision oncology. The findings from this study support the development of targeted therapeutic approaches focused on these genetic variants, potentially leading to improved patient outcomes. Additionally, understanding these genetic profiles could aid in identifying high-risk individuals who would benefit from enhanced surveillance and early intervention. While initial simple comparative analyses suggested that *TP53* alterations might merely represent common background variants across the cohorts, deeper clinicopathological evaluation using logistic regression modeling revealed a more complex dynamic. Through rigorous multivariate statistical analysis, this study establishes that *TP53* status plays a critical, independent role in tumor progression. Specifically, wild-type *TP53* preservation is a significant protective factor against metastasis (adjusted OR = 0.083, P = 0.045), an effect that remains independent of patient demographic factors such as age and sex. Furthermore, binary logistic regression identified that larger primary tumor size is an independent clinical predictor of metastatic potential (adjusted OR = 1.42, P = 0.029). It is important to note the methodological rigor applied to these models. To ensure robust statistical validity and prevent mathematical overfitting due to sample size limitations within the clinical subgroups (n = 35 for metastasis analysis), inherently linked tautological variables, namely tumor stage and metastatic status, were evaluated independently. Furthermore, genetic markers lacking mutational variance in this specific cohort (such as *IDH1* and *KRAS* codon 12) were systematically excluded from the regression models. Ultimately, these statistically significant associations highlight a unique mutational landscape in which specific genetic drivers dictate metastatic potential, a finding that warrants further functional validation in larger, multicenter iCCA cohorts.

In conclusion, this study provides important insights into the molecular genetic landscape of iCCA in a Thai population. *CREBBP* and *KRAS* codon 13 polymorphisms were significantly enriched in iCCA and likely represent key germline

variants contributing to tumorigenesis. These findings emphasize the need for further research to translate these genetic insights into clinical practice, ultimately improving prevention, diagnosis, treatment, and prognosis of iCCA.

### 4.4. Study limitations

Several limitations of this study should be acknowledged. First, the relatively modest sample size, particularly in subgroup analyses involving metastatic status, tumor size, tumor markers, and evaluation of the *KEAP1-NFE2L2* polymorphism, limited statistical power to detect moderate associations and may have contributed to wide confidence intervals. This constraint is common in studies of iCCA, a relatively rare malignancy, and it limits the ability to draw definitive conclusions about genotype–phenotype correlations. Second, the genetic analysis was limited to a targeted panel of selected candidate genes and polymorphisms. Although these genes represent biologically relevant pathways involved in tumor suppression, oncogenic signaling, and oxidative stress regulation, this approach may have overlooked additional germline variants, structural variants, or epigenetic alterations that contribute to iCCA pathogenesis. Comprehensive genomic approaches would provide a more complete representation of the molecular landscape. Third, functional validation studies were not performed to confirm the biological effects of identified germline variants. While statistical associations strongly support the involvement of *CREBBP* and *KRAS* codon 13 polymorphisms in iCCA, their mechanistic roles in cholangiocyte transformation, tumor progression, and therapeutic response require experimental confirmation. Furthermore, it is important to note that this study is observational in nature and relies on associative analysis. The identification of these variants does not establish a causal role in tumor development or progression. Future functional studies and experimental validation are required to determine whether these candidate variants act as true biological drivers. A notable limitation of this study, and of liquid biopsies in general, is that cell-free DNA extracted from serum represents a systemic genomic pool shed by various tissues undergoing apoptosis or necrosis throughout the body. Consequently, there is a risk that the detected variants may not originate exclusively from the primary tumor of interest but could theoretically arise from undetected concurrent diseases in other organs, benign neoplasms, or clonal hematopoiesis of indeterminate potential (CHIP). While standard somatic variant calling cannot definitively determine the tissue of origin, future studies can overcome this limitation by integrating multi-omic approaches. For instance, matched sequencing of peripheral blood mononuclear cells (PBMCs) can be utilized to filter out CHIP-derived variants. Furthermore, incorporating cfDNA epigenetic analysis, specifically DNA methylation profiling or fragmentomics, offers a promising solution, as methylation signatures are highly tissue-specific and can accurately pinpoint the anatomical origin of circulating DNA, thereby distinguishing tumor-specific signals from systemic biological noise. Finally, a *post hoc* power analysis indicates that our small sample size limited our statistical power to detect moderate associations and may have contributed to wide confidence intervals. This constraint is common in studies of iCCA, a relatively rare malignancy, and it limits the ability to draw definitive conclusions about genotype-phenotype correlations. For example, regression models featuring *CDKN2A* yielded an adjusted OR of 0.000 and a P-value of 1.000. This indicates quasi-complete separation in the data, likely because every patient with a *CDKN2A* mutation had the same metastasis outcome, highlighting how the sample size for specific rare variants limits the interpretability of their predictive value.

### Supporting information

**S1 Table. Clinical and genetic data of patients with cholangiocarcinoma and *Opisthorchis viverrini* infection and healthy subjects.**
(DOCX)

### Acknowledgments

We thank the staff of the Drug Discovery and Development Center at Thammasat University for their technical support.

## Author contributions

**Conceptualization:** Wanna Chaijaroenkul, Kesara Na-Bangchang.

**Data curation:** Rehab Osman Taha, Wanna Chaijaroenkul.

**Formal analysis:** Papichaya Phompradit.

**Funding acquisition:** Kesara Na-Bangchang.

**Investigation:** Rehab Osman Taha, Wanna Chaijaroenkul, Papichaya Phompradit.

**Methodology:** Wanna Chaijaroenkul, Papichaya Phompradit.

**Supervision:** Wanna Chaijaroenkul, Kesara Na-Bangchang.

**Validation:** Wanna Chaijaroenkul.

**Visualization:** Kesara Na-Bangchang.

**Writing – original draft:** Rehab Osman Taha, Wanna Chaijaroenkul.

**Writing – review & editing:** Kesara Na-Bangchang.

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
