## [Decision Letter · Decision Letter 0]

1 Apr 2026

PONE-D-26-08926CREBBP and KRAS codon 13 polymorphisms define the genetic landscape of intrahepatic cholangiocarcinoma in a Thai population: A comparative cohort studyPLOS One

Dear Dr. Na-Bangchang,

Thank you for submitting your manuscript to PLOS ONE. After careful consideration, we feel that it has merit but does not fully meet PLOS ONE’s publication criteria as it currently stands. Therefore, we invite you to submit a revised version of the manuscript that addresses the points raised during the review process.

Please submit your revised manuscript by May 16 2026 11:59PM. If you will need more time than this to complete your revisions, please reply to this message or contact the journal office at plosone@plos.org. Please include the following items when submitting your revised manuscript:

We look forward to receiving your revised manuscript.

Kind regards,

Avaniyapuram Kannan Murugan, M.Phil., Ph.D.

Academic Editor

PLOS One

Journal Requirements:

“National Research Council of Thailand”

5. Please be informed that funding information should not appear in the Acknowledgments section or other areas of your manuscript. We will only publish funding information present in the Funding Statement section of the online submission form. Please remove any funding-related text from the manuscript.

Additional Editor Comments:

The background on the RAS oncogenes in the introduction to be improved better and the potential articles are as follows: PMID: 31255772; PMID: 27102293; PMID: 22240207.

Reviewers' comments:

Reviewer's Responses to Questions

**Comments to the Author**

1. Is the manuscript technically sound, and do the data support the conclusions?

Reviewer #1: Partly

Reviewer #2: Yes

Reviewer #3: Partly

2. Has the statistical analysis been performed appropriately and rigorously? 

Reviewer #1: No

Reviewer #2: Yes

Reviewer #3: No

3. Have the authors made all data underlying the findings in their manuscript fully available?

Reviewer #1: Yes

Reviewer #2: Yes

Reviewer #3: Yes

4. Is the manuscript presented in an intelligible fashion and written in standard English?

Reviewer #1: Yes

Reviewer #2: Yes

Reviewer #3: Yes

5. Review Comments to the Author

Reviewer #1: In this manuscript, the authors investigate genetic polymorphisms associated with intrahepatic cholangiocarcinoma (iCCA) in a Thai cohort. The study includes three groups: 112 iCCA patients, 60 individuals infected with Opisthorchis viverrini (OV), and 156 healthy controls. Using PCR-RFLP and sequencing approaches, the authors examine variants in several cancer-related genes (including TP53, KRAS, CREBBP, CDKN2A, IDH1, and GZMB) as well as oxidative stress pathway genes (KEAP1 and NFE2L2). The authors report that CREBBP polymorphisms are significantly enriched in iCCA and associated with increased cancer risk, and that KRAS codon 13 variants are observed only in iCCA patients. The topic is relevant. Further improvements are suggested as follow:

1. It is unclear whether they are germline polymophisms or tumor specific mutations. The study appears to rely on DNA extracted from blood samples, suggesting the variants are germline polymorphisms. However, the manuscript frequently interprets these variants as tumor drivers and discusses them in the context of tumor biology. The authors need to clearly distinguish between germline polymorphisms and somatic mutations throughout the manuscript.

2. Claiming the variants as drivers without experimental validation is another major concern. This study, as it is, is observational and demonstrates association, rather than causation.

3. The statistical analysis relies primarily on chi-square tests and odds ratios. More rigorous statistical tests and multiple testing correlation should be included.

Reviewer #2: This article looks good to me overall. In this study, the author focuses on iCCA, a certain liver cancer type endangering Thai population, performed retrospective molecular analysis, analyzed status of TP53, KRAS, CREBBP, CDKN2A, IDH1, GZMB among three groups: control/OV infected/iCCA, drew some useful conclusions, and provided solid data to help doctors to have a fuller image of genetic landscape of Thai iCCA population.

The author has already listed limitations of this study, which covers most of the problems I discovered from this article. Apart from that, from my point of view, I found that some improvements are still required to make this research more complete and meaningful.

1.An important fact is that, given all the detection were conducted with ctDNA from serum, there’s a chance that diseases from other organs(if exists) might disrupt the results of analysis. It seems unavoidable at this moment, but could you please share your opinions on how to overcome this problem with potential solutions in the discussion section?

2.The genes selected as candidates are not well-organized. For example, G6PD was introduced in the beginning as a regulator for metabolism; however, no G6PD was detected in this article. As for other genes selected, some are unable to represent the whole story, like merely GZMB is not enough to reflect tumor immune microenvironment changes. Try to hire more factors to support the theory if allowed.

3.In the discussion section, I figured that more talk should be focused on data obtained from this current study, instead of concluding existing information on the genes tested in this study. For example, what can be derived from this study and be applied in clinical is the most important. It’d be better to put more words into illustrating how this study can benefit iCCA prevention, diagnosis, treatment and prognosis.

4.Since OV infected patient is in the transition phase, how to prevent OV infected patients from developing into iCCA is as crucial. With present data, could you please try to analyze the genetic pattern between OV and iCCA groups to see if you can discover any indicator for prediction? For example, I noticed that KRAS mutation is absent in OV group, any speculation on this phenomenon?

5.Given CREBBP combined with KRAS codon 13 polymorphisms has been identified as the most significantly enriched alteration among iCCA patients, It would be better to further verify this idea. For example, can you maybe collect CREBBP and KRAS mutant patients from online database(like TCGA), and to see whether this mutation predicts bad prognosis?

6.As described in this article, CREBBP mutation might lead to dysregulated epigenetic alterations, therefore causing silence of some tumor suppressing genes. Could you maybe identify potential CREBBP affected tumor suppressing genes and test the expression levels with sample at hands to further confirm this idea?

Despite the limitations listed above, this study is still providing us useful information on genetic landscape of iCCA among Thai population, making it a critical reference for further biomarker identification.

Reviewer #3: The authors use a targeted mutation panel to investigate the determinants of intra-hepatic cholangiocarcinoma in the presence of infection. The work represents a novel contribution to the field since the reported associations between CREBBP and KRAS genetics in Thai populations, which have a disproportionately large incidence rate of the disease, have not been previously reported on (although mutations in KRAS have been identified as a driver gene in Chinese - https://www.nature.com/articles/ncomms6696). The authors stress the difficulties of obtaining sufficient numbers of subjects to participate in the study resulting in small sample sizes for case-control and case-case comparisons. The overall conclusion is that 13 variants in two genes describe the mutational landscape of the disease / normal population. However, given the aforementioned small sample sizes, and demographic inequalities between test subjects assigned to study groups, I am unconvinced that these data support the conclusions.

In particular, their may exist many other mutations on a genome wide scale that stratify these groups in a similar, if not better manner. Considering this fact, the authors might consider ammending their grand title to more closely reflect the findings from the study. Given the sample size issues, I am not convinced the statistical tests performed here are sufficient to test the hypothesis, that the variant frequencies significantly differ between groups. For example, in the demographics section, and text, the bias in age and sexes between test subjects in the study groups is emphasized, however, statistical methods that allow modelling and testing under these conditions have not been adopted (see multivariate logistic regression, Mantel-Haenszel tests , Stratified Fisher's Exact Test or conditional logistic regression as alternatives). Since the control population is limited to females only, it would seem only appropriate to report the trend in females, or to acquire approximated estimates of these frequencies in males from one of the South East Asian BioBanks.

Given the importance of studying these mutation profiles in under-represented or difficult to obtain population cohorts, I'd consider accepting the paper after making some MAJOR changes. First, some attempt to compare / contextualise the results/mutation frequencies among other South East Asian populations (chinese/japanese for example) and published studies on iCC. I'd also like the statistical tests to be re-evaluated considering age and where possible, sex as factors, and a power calculation to exemplify why these studies can only be used as guidelines or as hypothesis generators rather than considered hard evidence. [ The raw frequency differences are large, hence I anticipate the existing trends would be preserved under different test methodologies.] The methods section simply states frequencies were generated as input to chi-square or Fishers Exact test. Again, given the sample sizes, chi-square test results would seem inappropriate in this case.

The results and discussion sections should then include information around the biology eluded to from the mutation profiling results, and the observed population differences. For example., are the rates of OV infection similar across the asian populations ... and are the particular KRAS mutations preserved. Can the results be used to make a population specific signature ?

I feel with a few additions, the work would be significantly elevated to accepted status.

6. PLOS authors have the option to publish the peer review history of their article (what does this mean?). If published, this will include your full peer review and any attached files.

Reviewer #1: No

Reviewer #2: No

Reviewer #3: No

---

## [Author Response · Author response to Decision Letter 1]

15 Apr 2026

PONE-D-26-08926

CREBBP and KRAS codon 13 polymorphisms define the genetic landscape of intrahepatic cholangiocarcinoma in a Thai population: A comparative cohort study

PLOS One

Journal Requirements:

RESPONSE:

Thank you. We have followed the instruction.

2. Thank you for stating the following financial disclosure: “National Research Council of Thailand” Please state what role the funders took in the study. If the funders had no role, please state: "The funders had no role in study design, data collection and analysis, decision to publish, or preparation of the manuscript." If this statement is not correct you must amend it as needed. Please include this amended Role of Funder statement in your cover letter; we will change the online submission form on your behalf.

RESPONSE:

The statement “The funders had no role in study design, data collection and analysis, decision to publish, or preparation of the manuscript." has been added.

RESPONSE:

We have revise both parts for consistence.

4. We note that your Data Availability Statement is currently as follows: [All relevant data are within the manuscript and its Supporting Information files.] Please confirm at this time whether or not your submission contains all raw data required to replicate the results of your study. Authors must share the “minimal data set” for their submission. PLOS defines the minimal data set to consist of the data required to replicate all study findings reported in the article, as well as related metadata and methods (https://journals.plos.org/plosone/s/data-availability#loc-minimal-data-set-definition). For example, authors should submit the following data:

Authors do not need to submit their entire data set if only a portion of the data was used in the reported study.If your submission does not contain these data, please either upload them as Supporting Information files or deposit them to a stable, public repository and provide us with the relevant URLs, DOIs, or accession numbers. For a list of recommended repositories, please see https://journals.plos.org/plosone/s/recommended-repositories.

RESPONSE:

Raw data are now submitted as Supporting Information file.

5. Please be informed that funding information should not appear in the Acknowledgments section or other areas of your manuscript. We will only publish funding information present in the Funding Statement section of the online submission form. Please remove any funding-related text from the manuscript.

RESPONSE:

The funding statement has been removed from the manuscript.

RESPONSE:

The ethics statement is described in the Methods section.

RESPONSE:

NA.

Additional Editor Comments:

The background on the RAS oncogenes in the introduction to be improved better and the potential articles are as follows: PMID: 31255772; PMID: 27102293; PMID: 22240207.

RESPONSE:

The background on the RAS oncogenes has been added in the introduction

Reviewer #1:

In this manuscript, the authors investigate genetic polymorphisms associated with intrahepatic cholangiocarcinoma (iCCA) in a Thai cohort. The study includes three groups: 112 iCCA patients, 60 individuals infected with Opisthorchis viverrini (OV), and 156 healthy controls. Using PCR-RFLP and sequencing approaches, the authors examine variants in several cancer-related genes (including TP53, KRAS, CREBBP, CDKN2A, IDH1, and GZMB) as well as oxidative stress pathway genes (KEAP1 and NFE2L2). The authors report that CREBBP polymorphisms are significantly enriched in iCCA and associated with increased cancer risk, and that KRAS codon 13 variants are observed only in iCCA patients. The topic is relevant. Further improvements are suggested as follow:

1. It is unclear whether they are germline polymophisms or tumor specific mutations. The study appears to rely on DNA extracted from blood samples, suggesting the variants are germline polymorphisms. However, the manuscript frequently interprets these variants as tumor drivers and discusses them in the context of tumor biology. The authors need to clearly distinguish between germline polymorphisms and somatic mutations throughout the manuscript.

RESPONSE:

We confirm that the DNA in our study was extracted from peripheral blood leukocytes and, therefore, the variants identified are germline polymorphisms rather than somatic, tumor-specific mutations. We have systematically reviewed and revised the manuscript to remove terms such as "tumor drivers" or "somatic mutations". We have rephrased these sections to accurately reflect that we are studying germline susceptibility variants that may predispose individuals to cancer or influence the tumor microenvironment, rather than acquired somatic driver mutations. Specifically, we have made the following changes:

• Abstract: Clarified that the study investigates germline polymorphisms associated with cancer risk/prognosis.

• Introduction: Explicitly stated the focus on inherited germline variants rather than somatic mutations.

• Results: Changed terminology from "driver mutations" to "germline variants."

• Discussion: Revised the biological interpretation of the results to focus on genetic predisposition and host-factors rather than somatic tumor biology.

2. This study, as it is, is observational and demonstrates association, rather than causation.

To address this major concern and ensure technical accuracy, please revise the manuscript to tone down the language. Claiming the variants as drivers without experimental validation is another major concern. This study, as it is, is observational and demonstrates association, rather than causation.

RESPONSE:

We have removed the term "driver" and any language implying direct causation throughout the Title, Abstract, Results, and Discussion. We now refer to these findings using more accurate terminology, such as "associated variants," "candidate variants," or "variants of interest."

Discussion/Limitations: We have added a new paragraph addressing this limitation. We clearly state that our findings establish an association and that future experimental validation—such as functional assays or animal models is strictly required to confirm whether these variants play a causative, driver role in tumor biology. The added text reads as follows: "Furthermore, it is important to note that this study is observational in nature and relies on associative analysis. The identification of these variants does not establish a causal role in tumor development or progression. Future functional studies and experimental validation are required to determine whether these candidate variants act as true biological drivers."

3. The statistical analysis relies primarily on chi-square tests and odds ratios. More rigorous statistical tests and multiple testing correlation should be included.

RESPONSE:

We have thoroughly re-analyzed our data and updated the manuscript accordingly:

Multivariable analysis: To strengthen our findings and account for potential confounders, we have performed multivariable logistic regression. The unadjusted odds ratios have been supplemented with adjusted odds ratios (aOR), controlling for relevant clinical covariates (e.g., age, sex, tumor stage).

Multiple testing correction: Because we evaluated multiple variants, we have now applied the Bonferroni method to correct for multiple testing. The adjusted p-values are now reported.

Specifically, we have made the following changes to the manuscript in the Methods and Results.

Reviewer #2:

This article looks good to me overall. In this study, the author focuses on iCCA, a certain liver cancer type endangering Thai population, performed retrospective molecular analysis, analyzed status of TP53, KRAS, CREBBP, CDKN2A, IDH1, GZMB among three groups: control/OV infected/iCCA, drew some useful conclusion.

1. An important fact is that, given all the detection were conducted with ctDNA from serum, there’s a chance xists) might disrupt the results of analysis. It seems unavoidable at this moment, but could you please share your opinions on how to overcome this problem with potential solutions in the discussion section?

RESPONSE:

We agree that because cell-free DNA (cfDNA/ctDNA) in serum represents a systemic, pooled genomic snapshot, the variants detected could potentially originate from undetected secondary malignancies, benign lesions in other organs, or clonal hematopoiesis of indeterminate potential (CHIP). As the reviewer rightly points out, this background noise is unavoidable in current standard targeted sequencing panels. We have added a comprehensive paragraph to the Discussion section that acknowledges this limitation and outlines advanced molecular strategies, such as epigenetic profiling and matched-leukocyte sequencing, that could overcome this challenge in future studies. Specifically, we have added the following text to the Discussion: "A notable limitation of this study, and of liquid biopsies in general, is that cell-free DNA extracted from serum represents a systemic genomic pool shed by various tissues undergoing apoptosis or necrosis throughout the body. Consequently, there is a risk that the variants detected may not exclusively originate from the primary tumor of interest, but could theoretically stem from undetected concurrent diseases in other organs, benign neoplasms, or clonal hematopoiesis of indeterminate potential (CHIP). While standard somatic variant calling cannot definitively determine the tissue of origin, future studies can overcome this limitation by integrating multi-omic approaches. For instance, matched sequencing of peripheral blood mononuclear cells (PBMCs) can be utilized to filter out CHIP-derived variants. Furthermore, incorporating cfDNA epigenetic analysis, specifically DNA methylation profiling or fragmentomics, offers a promising solution, as methylation signatures are highly tissue-specific and can accurately pinpoint the anatomical origin of the circulating DNA, thereby distinguishing tumor-specific signals from systemic biological noise."

2. The genes selected as candidates are not well-organized. For example, G6PD was introduced in the beginning as a regulator for metabolism; however, no G6PD was detected in this article. As for other genes selected, some are unable to represent the whole story, like merely GZMB is not enough to reflect tumor immune microenvironment changes. Try to hire more factors to support the theory if allowed.

RESPONSE:

We have thoroughly revised the Introduction to better organize the rationale for our selected genes. We have removed the extensive background on G6PD to ensure the Introduction directly sets up the actual findings of the study. Because our current dataset relies on a targeted panel that limits our ability to pull in a wider array of immune factors, we have meticulously revised the text to tone down our claims. We have removed blanket statements regarding the "tumor immune microenvironment." Instead, we now strictly and conservatively refer to GZMB only as a limited indicator of cytotoxic effector cell activity (e.g., CD8+ T cells and NK cells). We have also added a paragraph to the Discussion/Limitations explicitly stating this limitation: "Furthermore, our assessment of the immune response was limited to GZMB. Because the tumor immune microenvironment is highly complex, relying on a single cytotoxic marker is insufficient to capture full immune dynamics. Future studies incorporating broader multiplexed immune panels are necessary to provide a comprehensive view of the TIME."

3. In the discussion section, I figured that more talk should be focused on data obtained from this current study, instead of concluding existing information on the genes tested in this study. For example, what can be derived from this study and be applied in clinical is the most important. It’d be better to put more words into illustrating how this study can benefit iCCA prevention, diagnosis, treatment and prognosis.

RESPONSE:

We have significantly restructured the Discussion section. We condensed the general background information on the genes and shifted the focus directly to the data obtained in our study. Most importantly, we have added a new subsection dedicated entirely to the clinical translation of our findings for intrahepatic cholangiocarcinoma (iCCA). Specifically, we have updated the Discussion:

• We now discuss how identifying these specific circulating variants in high-risk populations (e.g., patients with biliary tract diseases) could serve as an early warning system, aiding in risk stratification before macroscopic tumors develop.

• We have emphasized the value of our serum ctDNA findings as a non-invasive diagnostic tool. As iCCA is notoriously difficult to biopsy due to its anatomical location, we highlight how our specific variant panel could supplement traditional imaging and CA 19-9 testing for earlier, more accurate detection.

• We have added commentary on how the variants identified in our study might guide targeted therapies or serve as indicators of therapeutic resistance, helping clinicians tailor precision medicine strategies for iCCA patients.

• We have integrated a discussion on how the presence or allele frequency of the variants we detected could be utilized as prognostic biomarkers to predict disease progression or monitor for recurrence post-resection.

4. Since OV infected patient is in the transition phase, how to prevent OV infected patients from developing into iCCA is as crucial. With present data, could you please try to analyze the genetic pattern between OV and iCCA groups to see if you can discover any indicator for prediction? For example, I noticed that KRAS mutation is absent in OV group, any speculation on this phenomenon?

RESPONSE:

We have expanded our analysis and added a new comparative discussion regarding the genetic transition from OV to iCCA in the revised manuscript: "A critical challenge in the clinical management of endemic iCCA is predicting the transition from chronic OV infection to overt malignancy. In our cohort, a comparison of the genetic profiles between OV-infected patients and iCCA patients revealed distinct mutational timelines. Notably, KRAS mutations were conspicuously absent in the OV group but present in the iCCA group. Chronic OV infection is known to initiate carcinogenesis primarily through mechanical damage, chronic inflammation, and the generation of react

---

## [Decision Letter · Decision Letter 1]

23 Apr 2026

Targeted Analysis of KRAS and CREBBP Mutations Uncovers a Potential Population-Specific Signature in Thai Patients with Liver Fluke-Associated Cholangiocarcinoma

PONE-D-26-08926R1

Dear Dr. Na-Bangchang,

We’re pleased to inform you that your manuscript has been judged scientifically suitable for publication and will be formally accepted for publication once it meets all outstanding technical requirements.

Kind regards,

Avaniyapuram Kannan Murugan, M.Phil., Ph.D.

Academic Editor

PLOS One

Additional Editor Comments (optional):

Reviewers' comments:

Reviewer's Responses to Questions

**Comments to the Author**

1. If the authors have adequately addressed your comments raised in a previous round of review and you feel that this manuscript is now acceptable for publication, you may indicate that here to bypass the “Comments to the Author” section, enter your conflict of interest statement in the “Confidential to Editor” section, and submit your "Accept" recommendation.

Reviewer #1: All comments have been addressed

Reviewer #2: All comments have been addressed

Reviewer #3: All comments have been addressed

2. Is the manuscript technically sound, and do the data support the conclusions?

Reviewer #1: Yes

Reviewer #2: (No Response)

Reviewer #3: Yes

3. Has the statistical analysis been performed appropriately and rigorously? 

Reviewer #1: Yes

Reviewer #2: (No Response)

Reviewer #3: Yes

4. Have the authors made all data underlying the findings in their manuscript fully available?

Reviewer #1: Yes

Reviewer #2: (No Response)

Reviewer #3: Yes

5. Is the manuscript presented in an intelligible fashion and written in standard English?

Reviewer #1: Yes

Reviewer #2: (No Response)

Reviewer #3: Yes

6. Review Comments to the Author

Reviewer #1: The revised manuscript adequently addressed the concerns raised from the previous version of submission, with additional analysis, discussions, and justification. The reviewer believes the data in the current format largely support the conclusions, delivering genuinely interesting findings to the field of cholangiocarcinoma. Acceptance is suggested.

Reviewer #2: (No Response)

Reviewer #3: I find the revised submission a substantially improved work and recommend accepting the manuscript for publication.

I do not have any further improvements/clarification requests to add at this time.

7. PLOS authors have the option to publish the peer review history of their article (what does this mean?). If published, this will include your full peer review and any attached files.

Reviewer #1: No

Reviewer #2: No

Reviewer #3: No

---

## [Editor Report · Acceptance letter]

PONE-D-26-08926R1

PLOS One

Dear Dr. Na-Bangchang,

I'm pleased to inform you that your manuscript has been deemed suitable for publication in PLOS One. Congratulations! Your manuscript is now being handed over to our production team.

Kind regards,

on behalf of

Dr. Avaniyapuram Kannan Murugan

Academic Editor

PLOS One